# Localized Linear Temporal Dynamics for Self-supervised Skeleton Action Recognition

## Abstract

Skeleton-based action recognition has achieved remarkable performance on large-scale benchmark datasets. Nevertheless, the scarcity of annotated skeleton action data poses a significant obstacle to the advancement of this field, which inspired the recent endeavors to explore self-supervised approaches. Among them, contrastive learning based methods have gained significant attention, wherein existing approaches primarily employ a spatial-temporal backbone to extract features from the action sequence and then conduct contrastive learning within the feature space. Skeleton action sequence itself is a highly complex dynamical system, yet existing methods overlook the higher-order temporal information of the sequence. In this work, we introduce Koopman Temporal Contrastive Learning (KTCL), a Koopman theory inspired contrastive learning framework, which focuses on the localized latent dynamics of the sequence by learning discriminative linear system dynamics. Given an action sequence, we first map it into a new space where the temporal evolution becomes linear. A dynamics-oriented contrastive loss is used to enforce the dynamics of positive (or negative) samples more similar (or dissimilar). To tackle the diverse dynamics across different action phases within one sequence, we further introduce segment-level localized linear dynamics, accompanied by a cross-matching mechanism for alignment. Additionally, a cross-order contrastive loss is proposed to further amplify the effect of contrast across features of different orders. Intensive experiments on four benchmark datasets show that the proposed methods achieve superior performance than competing methods.

## 1 Introduction

Human action recognition is a fundamental task in the field of computer vision with various applications, including video surveillance, autonomous driving, human-robot interaction, etc. In recent years, skeleton-based action recognition has become increasingly popular owing to the robustness of the skeleton modality under visual variations. Although supervised methods (Qian et al., 2022; Chen et al., 2021; Xu et al., 2022; Lee et al., 2022; Yang et al., 2022; Chi et al., 2022; Cai et al., 2023; Zhou et al., 2023a) have achieved remarkable performance on the large-scale datasets, they still rely heavily on full-supervision and require tremendous manual annotation. The full exploitation of a massive amount of unlabeled data has become a crucial concern.

To address this issue, self-supervised methods have been introduced to reduce the reliance on labeled data. Early works (Yang et al., 2023; Paoletti et al., 2022; Chen et al., 2022a; Su et al., 2020; 2021b; Zheng et al., 2018) adopt a reconstruction-based encoder-decoder framework to learn the latent representation in an unsupervised manner, which is subsequently employed for further downstream tasks like recognition and retrieval. Recently, contrastive learning (Zhou et al., 2023b; Dong et al., 2022; Shah et al., 2023; Lin et al., 2023; Franco et al., 2023; Mao et al., 2022; Zhang et al., 2022a;b; Guo et al., 2022) is widely used to learn discriminative skeleton action representation in a self-supervised fashion. These methods apply data augmentation techniques on skeleton action sequences to obtain positive and negative pairs, and then employ various spatial-temporal backbones to extract features from the action sequence. The learning process is supervised using a similarity-based contrastive loss in the feature space. However, skeleton action sequence has complex underlying dynamics, constituting a highly nonlinear dynamic system in nature. Existing methods employ max/average pooling to aggregate temporal information, thereby conducting contrastive learning on the first-order features of the sequence while neglecting the evolution pattern of the dynamical

system. Instead, our objective is to perform contrastive learning directly on the sequence dynamics, aiming to learn discriminative temporal evolution patterns that capture rich high-order information.

Directly comparing the evolution patterns of different dynamical systems is challenging, since the dynamic of the action sequence is highly nonlinear. We resort to Koopman theory (Koopman, 1931), a method from classical control theory for tackling non-linear dynamical systems. The Koopman operator transforms a non-linear dynamical system into a linear one, enabling mathematical analysis and manipulation of the system based on linear spectral theory. The Koopman operator can be obtained via either traditional methods like DMD, or deep learning based method which enables efficient end-to-end optimization on large datasets. Deep Koopman methods are widely used in sequential tasks, such as sequence prediction (Azencot et al., 2020; Mitjans et al., 2022), sequential disentanglement(Berman et al., 2023), sequence classification (Zhang et al., 2021; Wang et al., 2023), and model predictive control (Morton et al., 2019; Han et al., 2021). However, few studies have examined the potential use of the Koopman method in temporal unsupervised contrastive learning.

This paper is the first work that utilizes Koopman theory to formulate a contrastive learning framework for temporal sequence which leverages segment-level localized linear dynamics. As shown in Figure 1, the key idea is to elevate the similarity of the temporal dynamics for positive sample pairs, while enhancing the dissimilarity of dynamics for negative sample pairs. In order to facilitate the comparison of non-linear dynamics among action sequences, Koopman theory is employed to linearize the dynamics, enabling the contrast of system dynamics through the linear dynamic matrix. A specially designed linear dynamics contrastive loss is utilized to facilitate the model in learning discriminative temporal evolution patterns while ensuring the linear evolution of the system. When dealing with skeleton action sequences, we observe that different phases of the action can possess different dynamics. For instance, the action 'shoot with gun' may involve several stages such as holding the gun, raising the gun, and firing the gun. Using a single global linear dynamics to represent all phases can potentially result in errors and inaccuracies. In light of this, we introduce localized linear dynamics, wherein every action feature sequence is subdivided into multiple segments, each characterized by its unique linear dynamics. A cross-matching mechanism is employed to align the action segments of two sequences with their respective localized dynamics. Apart from the high-order dynamical features, we also utilize existing spatial-temporal backbone to extract first-order features and incorporate them into the architecture to make the learned representation more comprehensive. Furthermore, based on the assumption that features from the same action sequence of both orders should possess analogous semantic attributes, we map them into a joint embedding space and propose a cross-order contrastive loss to ensure the proximity between the high-order dynamical features and the low-order spatio-temporal features of positive pairs, while distinctly separating the feature representations of negative pairs. Our contributions can be summarized as follows:

- We propose Koopman Temporal Contrastive Learning (KTCL), the first Koopman-based temporal contrastive learning framework that leverages segment-level localized linear dynamics to acquire discriminative temporal evolution patterns for contrastive learning. A cross-matching mechanism is employed to align the segments for the proposed linear dynamics contrastive loss. The model further incorporates first-order information using existing backbone encoder, and adopts a cross-order contrastive loss to enhance the contrast in the joint embedding space.
- Comprehensive experiments on four benchmark datasets (NTU RGB+D, NTU RGB+D 120, PKUMMD-I and II datasets) under various settings demonstrate that KTCL achieves state-of-the-art performance in most cases, which validates the effectiveness of KTCL.

## 2 RELATED WORK

**Self-supervised Skeleton Action Recognition**. Self-supervised skeleton action recognition has attracted increasing attention in the past few years. Existing works can be mainly divided into two categories: contrastive-based methods and reconstruction-based methods. Contrastive-based methods (Zhou et al., 2023b; Dong et al., 2022; Shah et al., 2023; Xu et al., 2023; Men et al., 2023; Lin et al., 2023; Franco et al., 2023; Mao et al., 2022; Chen et al., 2022b; Zhang et al., 2022a; Lin et al., 2020; Li et al., 2021; Su et al., 2021a; Thoker et al., 2021; Xu et al., 2021a; Zhang et al., 2022b; Xu et al., 2021b; Guo et al., 2022; Wang et al., 2022) have emerged as the prominent strategy for self-supervised action recognition, which applies various contrastive losses to acquire more discriminative features. For instance, Dong et al. (2022) designed a hierarchical contrast framework

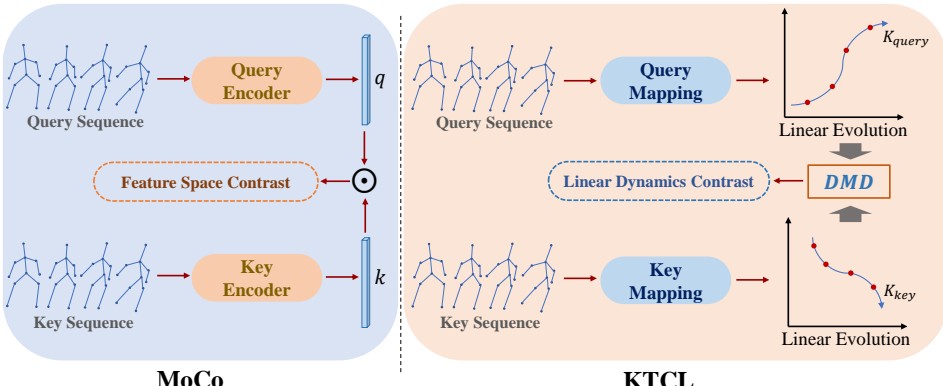

Figure 1: **Left**: existing MoCo-based temporal contrastive methods use a spatial-temporal encoder to extract the features of the action sequence, and then use inner-product to measure the similarity in the feature space for contrast. **Right**: our proposed Koopman temporal contrastive learning emphasizes the evolution pattern of the temporal sequence, and conducts contrastive learning on the localized linear dynamics.

by exploiting multiple-level representation and employing domain contrastive loss. Reconstruction-based methods (Yang et al., 2023; Paoletti et al., 2022; Chen et al., 2022a; Su et al., 2020; 2021b; Zheng et al., 2018; Xu et al., 2021b) usually adopt encoder-decoder structure to reconstruct masked input skeleton sequence. The learned latent representation can be used for further recognition tasks.

**Deep Koopman Learning**. Koopman theory (Koopman, 1931) is a longstanding method in the field of dynamical system analysis. To obtain Koopman operator, data-driven approaches are favorable when dealing with large-scale datasets, and many recent works (Lusch et al., 2018; Lee & Carlberg, 2020; Takeishi et al., 2017; Iwata & Kawahara, 2020; Johnson et al., 2022; Balakrishnan & Upadhyay, 2021) focus on the combination of deep learning and Koopman methods by employing neural networks as the lifting function of the system. As sequences are essentially the discrete form of dynamical systems in nature, Koopman methods can be readily applied to various sequential tasks such as prediction, disentanglement, and recognition (Azencot et al., 2020; Morton et al., 2019; Han et al., 2021; Mitjans et al., 2022; Berman et al., 2023; Zhang et al., 2021). Azencot et al. (2020) construct a Koopman-based auto-encoder structure by introducing forward and backward Koopman operators and a consistency loss, which can be applied for sequence prediction. Zhang et al. (2021) use Koopman linear dynamics to represent different walking patterns and construct a framework for gait recognition. Wang et al. (2023) use parameterized Koopman pooling to substitute average pooling for supervised skeleton action recognition. Note that the Koopman matrices in both Zhang et al. (2021) and Wang et al. (2023) are class-specific parameters and need to be learned from data.

## 3 PRELIMINARIES

### 3.1 BASICS OF KOOPMAN THEORY

We first present a brief review of the basics of Koopman theory, a mathematical framework for analyzing nonlinear dynamical systems. Given a discrete-time dynamical system,

$$y_{t+1} = F(y_t), \quad Y \in \mathbb{R}^d. \tag{1}$$

The real-valued observable function $\phi$ is defined on states $y_t$ as $x_t = \phi(y_t)$. The Koopman operator $\mathcal{K}$ (whose finite-dimensional approximation is denoted as $\mathbf{K}$) acts on the observable $\phi$, such that the resulting dynamics in the subspace is linearly evolving:

$$x_{t+1} = \phi(y_{t+1}) = \phi(F(y_t)) = \mathbf{K}\phi(y_t) = \mathbf{K}x_t. \tag{2}$$

Given a sequence of state representations $\mathbf{X} = \{x_1, x_2, \cdots, x_T\}$, one popular way to obtain the finite-dimensional approximation of the Koopman operator is through Dynamic Mode Decomposition (DMD) (Schmid, 2010; Williams et al., 2015; Brunton & Kutz, 2019), which is formulated as:

$$\mathbf{K} = \arg\min_{\mathbf{K}} \|\mathbf{X}_{2:T} - \mathbf{K}\mathbf{X}_{1:T-1}\|_F = \mathbf{X}_{2:T}\mathbf{X}_{1:T-1}^\dagger, \tag{3}$$

where $\|\cdot\|_F$ denotes the Frobenius norm and $\dagger$ is matrix pseudo-inverse. $\mathbf{X}_{i:j}$ stands for $[x_i, \cdots, x_j]$.

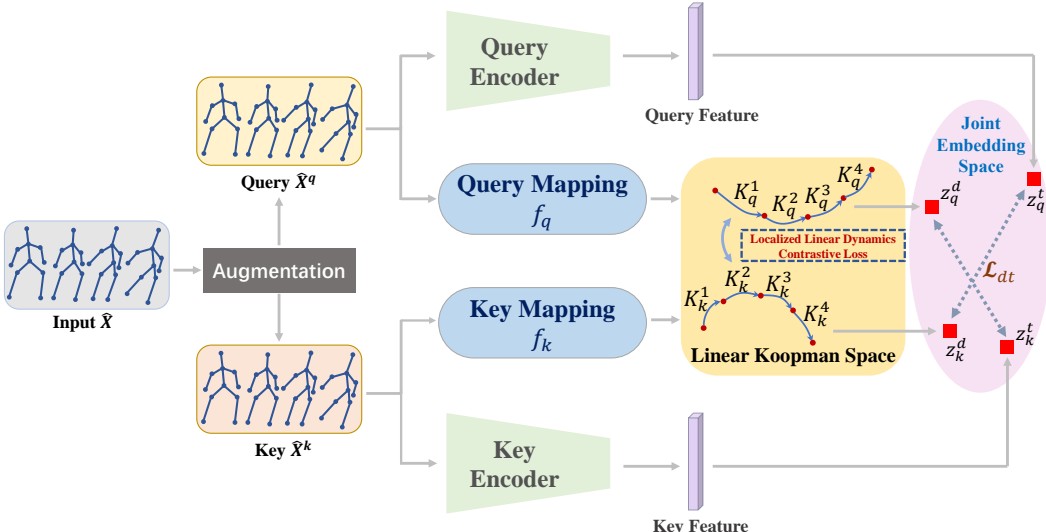

Figure 2: **Overall architecture of our proposed model.** Data augmentation is first applied to acquire query and key data, which serve as the positive sample pair. The sequence is then mapped into the linear Koopman space where the system evolution is linearized on segment-level and governed by the matrices $\mathbf{K}_q$ and $\mathbf{K}_k$. The localized linear dynamics contrastive loss is employed to conduct contrast learning in the Koopman space. Moreover, a spatio-temporal encoder is utilized to extract first-order features, which are subsequently mapped to the joint embedding space along with the dynamics matrices where cross-order contrastive loss $\mathcal{L}_{dt}/\mathcal{L}_{ds}$ is applied. $z^d, z^t$ denote the embeddings of Koopman matrix and first-order features in the joint embedding space respectively. Note that Koopman matrices are computed using DMD rather than being learned from data.

## 3.2 MoCo for Self-supervised Skeleton Action Recognition

In recent years, the MoCo-based framework (Chen et al., 2020; He et al., 2020) has gained widespread use in self-supervised skeleton action recognition. As our method is built upon the basic structure of MoCo, we will provide a brief review of the key components of the framework.

**Data Augmentations.** Suppose the input 3-D skeleton action sequence has the shape $\hat{\mathbf{X}} \in \mathbb{R}^{T \times 3V}$, where $T$ denotes sequence length and $V$ denotes the total number of joints, data augmentation is applied to obtain two different augmented sequence: query data $\hat{\mathbf{X}}^q$ and key data $\hat{\mathbf{X}}^k$. $(\hat{\mathbf{X}}^q, \hat{\mathbf{X}}^k)$ is regarded as the positive pair for contrastive learning.

**Encoders.** A query encoder $f_q$ and a key encoder $f_k$ are employed to extract spatial-temporal features $z_q, z_k$ of skeleton sequences $\hat{\mathbf{X}}^q$ and $\hat{\mathbf{X}}^k$ respectively. During the training process, only the parameters of the query encoder $f_q$ participate in gradient descent updates. The parameters of the key encoder $f_k$ are updated via the momentum update mechanism $\theta_k \leftarrow m\theta_k + (1-m)\theta_q$, where $\theta_k$ and $\theta_q$ are the parameters of $f_k$ and $f_q$, and $m$ is the momentum coefficient.

**Contrastive loss.** To acquire negative samples, a queue $M = \{m_i\}_{i=1}^r$ is constructed to store previous keys features $f_k(\hat{\mathbf{X}}^k)$ from recent iterations which serves as the memory bank. The similarity between the embeddings is measured as the normalized dot product, and the resulting contrastive loss function takes the form of InfoNCE (Oord et al., 2018) loss:

$$\mathcal{L} = -\log \frac{\exp(z_q \cdot z_k / \tau)}{\exp(z_q \cdot z_k / \tau) + \sum_{i=1}^r \exp(z_q \cdot m^i / \tau)}, \tag{4}$$

where $\tau$ is the temperature hyperparameter and $r$ is the size of memory bank.

## 4 The Proposed Method

Most existing contrast-based self-supervised skeleton action recognition methods follow the framework of MoCo described above. However, the InfoNCE loss only considers the similarity of extracted

first-order features, which ignores the complex underlying dynamics of the sequence. In this section, we present Koopman Temporal Contrastive Learning (KTCL), a localized linear dynamics-based contrastive learning framework inspired by Koopman theory. The overall architecture is shown in Figure 2, and the details of the model will be elaborated below.

## 4.1 KOOPMAN TEMPORAL CONTRASTIVE LEARNING

Given input 3-D skeleton action sequence $\hat{\mathbf{X}} \in \mathbb{R}^{T \times 3V}$, we first apply data augmentation techniques to transform $\hat{\mathbf{X}}$ into two augmented sequences: query data $\hat{\mathbf{X}}^q$ and key data $\hat{\mathbf{X}}^k$, which serve as the positive sample pairs. The augmented sequences are then mapped into the embedding space respectively by the mapping function, $i.e.$, $\mathbf{X}^q = f_q(\hat{\mathbf{X}}^q) \in \mathbb{R}^{T \times C}$, $\mathbf{X}^k = f_k(\hat{\mathbf{X}}^k) \in \mathbb{R}^{T \times C}$, where $C$ is the embedding dimension. In MoCo-based frameworks, embedding contrast is achieved through the inner-product based similarity of the embeddings. Such practice ignores the temporal evolution of the sequence. Instead, our primary objective is to perform contrastive learning on the temporal dynamics of $\mathbf{X}^q$ and $\mathbf{X}^k$, whereas the key challenge lies in effectively comparing the dynamics of the sequences. The crucial inspiration here is to aim for linearity in the evolution of sequences within the embedding space, enabling direct comparison of their temporal dynamics:

$$x_{t+1}^q = \mathbf{K}_q x_t^q, \quad x_{t+1}^k = \mathbf{K}_k x_t^k, \tag{5}$$

where $t = 1, 2, \cdots, T-1$. The linear transition matrices $\mathbf{K}_q$, $\mathbf{K}_k$ determine the temporal dynamics of the respective sequence, thus allowing the similarity measurement of the temporal dynamics through cross-sequence 1-step linear fitting error, denoted as:

$$\|\mathbf{X}_{2:T}^q - \mathbf{K}_k \mathbf{X}_{1:T-1}^q\|_F. \tag{6}$$

$\mathbf{K}_k$ is derived via the Dynamics Mode Decomposition (DMD) given key feature sequence $\mathbf{X}^k$:

$$\mathbf{K}_k = \mathbf{X}_{2:T}^k (\mathbf{X}_{1:T-1}^k)^\dagger, \tag{7}$$

where $\dagger$ denotes the pseudo-inverse. In an effort to learn discriminative temporal evolution patterns, the primary goal is to ensure similarity in the dynamics pattern of positive pairs and divergence in the dynamics pattern of negative pairs. To achieve this, we proposed the Linear Dynamics Contrastive (LDC) loss, which can be formulated as follows:

$$\mathcal{L}_{ldc} = -\log \frac{\exp(-\|\mathbf{X}_{2:T}^q - \mathbf{K}_k \mathbf{X}_{1:T-1}^q\|_F)}{\exp(-\|\mathbf{X}_{2:T}^q - \mathbf{K}_k \mathbf{X}_{1:T-1}^q\|_F) + \sum_{j=1}^r \exp(-\|\mathbf{X}_{2:T}^q - \mathbf{K}_j^n \mathbf{X}_{1:T-1}^q\|_F)}, \tag{8}$$

where $M_K = \{\mathbf{K}_j^n\}_{j=1}^r$ is a queue constructed to store previously calculated linear dynamics matrices, which are used as negative samples. Note that under such formulation, the cross-sequence linear fitting error $\|\mathbf{X}_{2:T}^q - \mathbf{K}_k \mathbf{X}_{1:T-1}^q\|_F$ for the positive pairs is minimized, therefore ensuring the linearity of the temporal evolution of $\mathbf{X}^q$ while simultaneously bringing the dynamics of $\mathbf{X}^q$ and $\mathbf{X}^k$ closer.

## 4.2 SEGMENT-LEVEL LOCALIZED LINEAR DYNAMICS

Global linearization of the entire sequence presents a formidable challenge. As shown in Figure 3(c), seeking global linearity often results in a relatively higher linear fitting error. In mathematical contexts, it is often more advantageous to pursue local linearization. For a skeleton action sequence, various phases of the action may exhibit varying dynamics. Moreover, as contrastive learning involves temporal augmentation to the action sequence, each action phase can manifest at any stage in the sequence. Thus, it becomes essential to investigate the local dynamics at segment level. To this end, we introduce segment-level linear dynamics as a complement to global dynamics, wherein each action sequence is subdivided into multiple segments, characterized by their respective localized linear dynamics.

Formally, the encoded feature sequence $\mathbf{X}^q$ and $\mathbf{X}^k$ are subdivide into $s$ segments: $[\mathbf{X}^{q,1}, \ldots \mathbf{X}^{q,s}]$ and $[\mathbf{X}^{k,1}, \ldots \mathbf{X}^{k,s}]$. Each sequence segment exhibits local linearity, $i.e.$

$$x_{t+1}^{q,j} = \mathbf{K}_{q,j} x_t^{q,j}, \quad x_{t+1}^{k,j} = \mathbf{K}_{k,j} x_t^{k,j}, \tag{9}$$

for $j$ in $1, \ldots s$.

Our objective is to maintain local linearity while enhancing the dynamics contrast between the segmented sequences. A naive way to achieve this goal is to align the segments $x_t^{q,j}, x_t^{k,j}$ sequentially, and then employ the aforementioned LDC loss for each segment pair. However, for two sequences belonging to the same action category, their action phases may not necessarily align according to the temporal order. Here, we introduce a cross-matching mechanism that matches the segments based on the linear dynamics distance(best-match), thereby enhancing the robustness of the method. The dynamics distance $D^{ih}$ is defined in the form of LDC loss:

$$D^{ih} = \log \frac{\exp(-\|\mathbf{X}_{2:T}^{q,i} - \mathbf{K}_{k,h}\mathbf{X}_{1:T-1}^{q,i}\|_F)}{\exp(-\|\mathbf{X}_{2:T}^{q,i} - \mathbf{K}_{k,h}\mathbf{X}_{1:T-1}^{q,i}\|_F) + \sum_{j=1}^{r} \exp(-\|\mathbf{X}_{2:T}^{q,i} - \mathbf{K}_j^n\mathbf{X}_{1:T-1}^q\|_F)}, \quad (10)$$

where $D^{ih}$ measure the distance between the $i^{th}$ segment of $\mathbf{X}^q$ and the $h^{th}$ segment of $\mathbf{X}^k$. For $\mathbf{X}^{q,i}$, its best-match segment in $\mathbf{X}^{k,h}$ is determined by applying a minimum operation along the $h$ dimension. Consequently, the Localized Linear Dynamics Contrastive (localized-LDC) loss minimizes the disparity in dynamics between each paired segment:

$$\mathcal{L}_{local} = -\sum_i \min_h D^{ih}. \quad (11)$$

Notice that $\mathcal{L}_{local}$ guarantees the linearity of each segment of $[\mathbf{X}^{q,1}, \ldots \mathbf{X}^{q,s}]$, while simultaneously fostering the similarity in the dynamics of each paired segment.

### 4.3 CROSS-ORDER CONTRASTIVE LOSS

Koopman temporal contrastive learning provides rich high-order information of the action sequence, thereby facilitating the model's ability to learn discriminative dynamical patterns. Nevertheless, it is acknowledged that the value of a dynamical system at a specific time is determined by both its dynamic equation and initial value. In order to obtain a holistic representation of an action sequence, it is essential to incorporate first-order features into the aforementioned linear dynamics. To this end, max/mean pooling based backbones derived from existing works are employed to extract spatiotemporal information, which serves as the first-order features of the sequence. In this work, we adopt HiCo (Dong et al., 2022), a hierarchical encoder network that encodes skeleton sequences at multiple granularities of both temporal and spatial domains. The acquired first-order features are subsequently mapped to the joint embedding space together with the global dynamics matrices $\mathbf{K}_q$ and $\mathbf{K}_k$. Our objective is to ensure that the global high-order dynamical features and the low-order spatio-temporal features of the positive pairs share analogous semantic meanings, thus exhibiting a considerable level of proximity in the joint embedding space. Meanwhile, the embedding belonging to negative pairs should be distinctly distant from each other.

Specifically, to obtain the dynamical representation, we utilize a simple MLP to encode the sequence's global dynamics matrix $\mathbf{K}_q, \mathbf{K}_k$, yielding the corresponding high-order dynamical embedding $z_q^d$ and $z_k^d$. The backbone encoder produces a temporal embedding and a spatial embedding, which are then mapped into the joint embedding space. This results in first-order temporal representations $z_q^t, z_k^t$ and spatial representations $z_q^s, z_k^s$. The cross-order contrastive loss is then formulated as:

$$\mathcal{L}_{dt} = -\log \frac{\exp(z_q^d \cdot z_k^t/\tau)}{\exp(z_q^d \cdot z_k^t/\tau) + \sum_{j=1}^r \exp(z_q^d \cdot m_j^t/\tau)} - \log \frac{\exp(z_q^t \cdot z_k^d/\tau)}{\exp(z_q^t \cdot z_k^d/\tau) + \sum_{j=1}^r \exp(z_q^t \cdot m_j^d/\tau)}, \quad (12)$$

$$\mathcal{L}_{ds} = -\log \frac{\exp(z_q^d \cdot z_k^s/\tau)}{\exp(z_q^d \cdot z_k^s/\tau) + \sum_{j=1}^r \exp(z_q^d \cdot m_j^s/\tau)} - \log \frac{\exp(z_q^s \cdot z_k^d/\tau)}{\exp(z_q^s \cdot z_k^d/\tau) + \sum_{j=1}^r \exp(z_q^s \cdot m_j^d/\tau)}, \quad (13)$$

where $\tau$ is the temperature hyperparameter, and $M_d = \{m_j^d\}_{j=1}^r, M_t = \{m_j^t\}_{j=1}^r, M_s = \{m_j^s\}_{j=1}^r$ are the negative sample bank. $\mathcal{L}_{ds}, \mathcal{L}_{dt}$ guide the learning in joint embedding space, aiming to enforce similar semantic attributes among embeddings of different orders for positive pairs. The local dynamics matrix of the sequence is also mapped into the embedding space using an MLP, resulting in the embedding denoted as $z^l$. The overall representation of the sequence is the concatenation of the above features: $z = \text{concat}(z^d, z^l, z^t, z^s)$.

The overall loss function can be computed as $\mathcal{L}_{overall} = \mathcal{L}_{ldc} + \mathcal{L}_{local} + \mathcal{L}_{dt} + \mathcal{L}_{ds} + \mathcal{L}_{backbone}$, where $\mathcal{L}_{backbone}$ is the loss function of first-order spatial-temporal backbone.

Table 1: Comparisons to the state-of-the-art methods for self-supervised skeleton-based action recognition on NTU-60, NTU-120, PKU-MMD I and II of joint modality. Performance with $^\dagger$ denotes the fusion result of the three modalities joint, bone, and motion. The best results are in bold.

| Methods | NTU-RGB+D 60 | | NTU-RGB+D 120 | | PKU-MMD I (%) | PKU-MMD II (%) |
| | X-Sub (%) | X-View (%) | X-Sub (%) | X-Set (%) | | |
|---|---|---|---|---|---|---|
| LongT GAN (Zheng et al., 2018) (AAAI'18) | 39.1 | 48.1 | - | - | 67.7 | 26.0 |
| P&C (Su et al., 2020) (CVPR'20) | 50.7 | 76.3 | 42.7 | 41.7 | - | 25.5 |
| MS$^2$L (Lin et al., 2020) (ACM MM'20) | 52.6 | - | - | - | 64.9 | 27.6 |
| SeBiReNet (Nie et al., 2020) (ECCV'20) | - | 79.7 | - | - | - | - |
| CrosSCLR (Li et al., 2021) (CVPR'21) | 72.9 | 79.9 | 67.9$^\dagger$ | 66.7$^\dagger$ | - | - |
| Colorization (Yang et al., 2021) (ICCV'21) | 75.2 | 83.1 | - | - | - | - |
| ISC (Thoker et al., 2021) (ACMMM'21) | 76.3 | 85.2 | 67.1 | 67.9 | 80.9 | 36.0 |
| AimCLR (Guo et al., 2022) (AAAI'22) | 74.3 | 79.7 | 63.4 | 63.4 | 83.4 | 38.5$^\dagger$ |
| CMD (Mao et al., 2022) (ECCV'22) | 79.8 | 86.9 | 70.3 | 71.5 | - | 43.0 |
| GL-Transformer (Kim et al., 2022) (ECCV'22) | 76.3 | 83.8 | 66.0 | 68.7 | - | - |
| HYSP (Franco et al., 2023) (ICLR'23) | 78.2 | 82.6 | 61.8 | 64.6 | 83.8 | - |
| PSTL (Zhou et al., 2023b) (AAAI'23) | 77.3 | 81.8 | 66.2 | 67.7 | 88.4 | 49.3 |
| HiCo (Dong et al., 2022) (AAAI'23) | 81.4$^\dagger$ | 88.8 | 73.7 | **74.5** | 89.4 | 54.7 |
| HiCLR (Zhang et al., 2022b) (AAAI'23) | 80.4$^\dagger$ | 85.5$^\dagger$ | 70.0$^\dagger$ | 70.4$^\dagger$ | - | - |
| ActCLR (Lin et al., 2023) (CVPR'23) | 80.9 | 86.7 | 69.0 | 70.5 | - | - |
| HaLP (Shah et al., 2023) (CVPR'23) | 79.7 | 86.8 | 71.1 | 72.2 | - | 43.5 |
| SkeAttnCLR (Hua et al., 2023) (IJCAI'23) | 80.3 | 86.1 | 66.3 | **74.5** | 87.3 | 52.9 |
| **KTCL (Ours)** | **82.4** | **89.4** | **74.4** | **74.5** | **89.9** | **55.5** |

## 5 EXPERIMENTS

### 5.1 DATASET DESCRIPTION

**NTU RGB+D** (Shahroudy et al., 2016) contains 56,880 skeleton action sequences from 60 classes, performed by 40 subjects. The sequences are captured from multiple viewpoints using three cameras. Two evaluation protocols are commonly used: cross-subject, where the 40 subjects are split into 20 training subjects and 20 testing subjects; cross-view, where the sequences from cameras 2 and 3 are used for training, and the rest are used for testing. **NTU RGB+D 120** (Liu et al., 2019) is an extension of NTU RGB+D, which contains 113945 skeleton action sequences of 120 classes, performed by 106 subjects. This dataset contains 32 setups (location and background). Two evaluation protocols are commonly used: cross-subject, where the 106 subjects are split into 53 training subjects and 53 testing subjects; cross-setup, where the sequences with even setup IDs are used for training, and the rest are used for testing. **PKU-MMD** (Liu et al., 2017) is a multi-modality human action dataset that contains about 20,000 action sequences from 51 classes. The dataset is split into two parts (PKU-MMD I and II), and the latter contains more challenging samples due to larger view variation.

### 5.2 EVALUATION PROTOCOL

**Linear Evaluation.** Under linear evaluation, after conducting self-supervised pretraining on unlabeled data, the model parameters are fixed. A linear classifier is then appended to the model and trained with supervision. The entire network except for the linear classification head remains frozen.

**KNN Evaluation.** KNN evaluation is a parameter-free protocol that assesses the quality of the learned representation. A K-Nearest Neighbour classifier is directly applied to the features derived from self-supervised pretraining. In accordance with prior works, the parameter $k$ in KNN is set to 1.

**Transfer Learning.** To assess the generalization ability of the model, we also conduct transfer learning, wherein the model is pretrained on the source dataset and then transferred to the target dataset. Specifically, a linear classifier is appended to the pretrained model, and both the classifier and the pretrained model are fine-tuned on the target dataset.

### 5.3 IMPLEMENTATION DETAILS

Contrastive learning requires data augmentation to obtain positive sample pairs. For a fair comparison, we follow previous works (Dong et al., 2022; Mao et al., 2022) and adopted 3 skeleton data augmentation techniques, namely pose augmentation, joint jittering and temporal crop-resize. The dimension of the linear dynamical space is set to 64(32) for global(local) dynamics respectively.

Table 2: KNN evaluation results on NTU 60 and 120.

| Methods | NTU-RGB+D 60 | | NTU-RGB+D 120 | |
|---|---|---|---|---|
| | X-Sub (%) | X-View (%) | X-Sub (%) | X-Set (%) |
| LongT GAN | 39.1 | 48.1 | 31.5 | 35.5 |
| ISC | 62.5 | 82.6 | 50.6 | 52.3 |
| HiCLR | 67.3 | 75.3 | - | - |
| HiCo | 66.9 | 84.3 | 56.3 | 59.1 |
| HaLP | 65.8 | 83.6 | 55.8 | 59.0 |
| SkeAttnCLR | **69.4** | 76.8 | 46.7 | 58.0 |
| KTCL (Ours) | 67.1 | **84.7** | **58.9** | **60.7** |

Table 3: Comparisons on transfer learning.

| Methods | Transfer to PKU-MMD II | |
|---|---|---|
| | PKU-MMD I | NTU-60 |
| LongT GAN | 43.6 | 44.8 |
| ISC | 45.1 | 45.9 |
| HYSP | 50.7 | 46.3 |
| HiCo | 53.4 | 56.3 |
| HaLP | - | 54.8 |
| ActCLR | - | 55.9 |
| KTCL (Ours) | **57.3** | **58.4** |

Table 4: Evaluation results on each data stream.

| Methods | joint | bone | motion | fusion |
|---|---|---|---|---|
| CMD | 70.3 | - | - | 74.7 |
| PSTL | 66.2 | 58.7 | 65.4 | 69.2 |
| HiCo | 73.7 | 72.9 | 68.9 | 75.9 |
| ActCLR | 69.0 | 67.8 | 68.3 | 74.3 |
| KTCL (Ours) | **74.4** | **74.2** | **69.1** | **76.6** |

Table 5: Ablation study.

| global | cross-order | local | alignment | Acc (%) |
|---|---|---|---|---|
| ✓ | | | cross | 73.6 |
| ✓ | ✓ | | cross | 74.0 |
| ✓ | | ✓ | cross | 73.9 |
| ✓ | ✓ | ✓ | order | 73.9 |
| ✓ | ✓ | ✓ | mean | 74.0 |
| ✓ | ✓ | ✓ | cross | **74.4** |

When calculating pseudo-inverse in DMD, the singular values that are below 0.09 are discarded to guarantee numerical stability during gradient backpropagation. The number of segments $s$ is set to 4.

## 5.4 RESULTS

Table 1 presents the experimental results of linear evaluation on NTU-60, NTU-120, PKU-MMD I and II datasets. Note that the experiments are conducted on joint modality, and the results for PKU-MMD are obtained using the cross-subject protocol. For comparison, we include recently proposed representative works from this year in the table, *e.g.* PSTL (Zhou et al., 2023b), HiCo (Dong et al., 2022), HYSP (Franco et al., 2023), ActCLR (Lin et al., 2023), HaLP (Shah et al., 2023), SkeAttnCLR (Hua et al., 2023), etc. In terms of HiCo's performance, we adopt HiCo-LSTM version since our sequence backbone is also LSTM. As shown, the proposed method achieved state-of-the-art performance in most cases. For instance, KTCL achieves a notable performance gain of 1.0% on NTU-60 X-sub and 0.8% on the challenging dataset PKU-MMD II. This indicates that incorporating high-order temporal dynamics contrast enables the model to learn discriminative evolution patterns in the Koopman linear space, thereby enhancing classification accuracy. To further demonstrate the effectiveness of KTCL across various data modalities, we present the linear evaluation results on joint, bone, and motion modality, along with the the results of 3-stream fusion in Table 4. The experiments are conducted on NTU 120 xsub. As demonstrated, KTCL consistently outperforms existing approaches across the modalities, thereby affirming the efficacy of the proposed method.

The results for KNN evaluation are presented in Table 2. Our model achieves state-of-the-art performance on NTU60 X-view, NTU120 X-Sub and X-Set. Notably, on the extensive NTU120 dataset, the model significantly surpasses the performance of existing methods. The superiority observed in parameter-free evaluation demonstrates the discrimination of the learned representation.

Table 3 presents the results for transfer learning evaluation. Following previous practice, the model is pretrained on the source datasets (PKU-MMD I and NTU-60) and then transferred to the target dataset (PKU-MMD II). The experiments are carried out on joint modality under the X-sub protocol, with the finetuning hyperparameters following a similar setting to the pretraining phase. As observed, KTCL significantly improves the performance by 3.9% on PKU-MMD I and 2.1% on NTU-60. The remarkable performance in transfer learning demonstrates the strong generalization ability of KTCL.

## 5.5 ABLATION STUDIES

In order to demonstrate the performance gain from each component, we conduct ablation studies on NTU-120 datasets. All experiments are carried out using the joint modality under X-sub protocol. In the table, 'global' stands for baseline models with only global dynamic features, 'local' stands

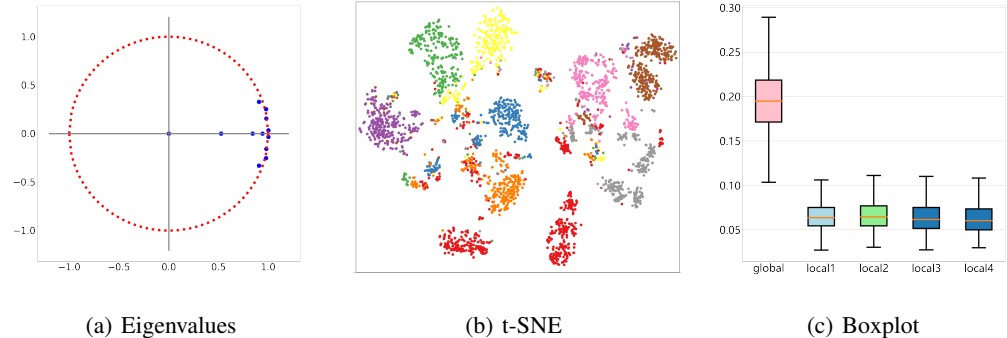

(a) Eigenvalues          (b) t-SNE          (c) Boxplot

Figure 3: **Visualizations.** (a) is the eigenvalues of global Koopman dynamics matrix on the complex plane. (b) is the t-SNE visualization of the learned representations. (c) is the box plot of the linear fitting error rate for local dynamics for all 4 segments and the global dynamics. Best viewed in color.

for models with localized dynamic features. 'Cross-order' denotes the inclusion of the cross-order contrastive loss. To showcase the effectiveness of our cross-matching mechanism, we additionally compare two other alignment techniques, namely 'order,' which aligns segments based on their temporal order; 'mean,' which changes the minimum operation in Equation 11 to mean operation. The results, presented in Table 5, demonstrate the effect of each component.

## 5.6 VISUALIZATIONS

We further conduct a series of visualizations to analyze the model. The spectrum of Koopman matrix holds significance in the Koopman methods as it dictates the system's dynamics patterns, including vital properties such as system stability (Erichson et al., 2019; Pan & Duraisamy, 2020; Azencot et al., 2020).In Figure 3(a), we visualize the eigenvalues of the global Koopman dynamics matrix $\mathbf{K}$ of a sample sequence. As observed, most of the eigenvalues are located near the unit circle. This indicates that the learned linear dynamic is stable and non-decaying, as an eigenvalue with a modulus greater than one would lead to an explosion while an eigenvalue with a modulus smaller than one would gradually decay over time. Figure 3(b) presents the t-SNE (Van der Maaten & Hinton, 2008) visualization results of the learned representation where each color denotes an action class, providing evidence that the model has learned discriminative features in the embedding space. To show the effect of linearization as well as the effectiveness of the localized linear dynamics, the linear fitting error rate is introduced, which is defined as the ratio of the absolute linear fitting error to the embedding norm: $error = \frac{\|\mathbf{X}_{2:T} - \mathbf{K}\mathbf{X}_{1:T-1}\|_F}{\|\mathbf{X}\|_F}$, where $\mathbf{K}$ is derived via DMD. We illustrate the box plot of the aforementioned error rate for the local dynamics of each segment and the global dynamics in Figure 3(c). As depicted, the median of the global dynamics error rate is approximately 19%, whereas the median of the local dynamics error rate falls within the range of 5% to 7%. This indicates that the dynamics in the embedding space are effectively linearized, with localized dynamics offering more precise linear fitting for various action phases.

## 6 CONCLUSION

In this work, we present Koopman Temporal Contrastive Learning, the first Koopman-inspired temporal contrastive learning framework aimed at learning discriminative localized linear temporal dynamics for self-supervised skeleton action recognition. By leveraging Koopman theory, the action sequence dynamic is linearized, facilitating a direct comparison of dynamical systems through the proposed linear dynamics contrastive loss. In order to address the diverse dynamics exhibited across various action phases within a single sequence, we introduce segment-level localized linear dynamics along with a cross-matching mechanism. Additionally, we integrate a first-order feature branch and introduce a cross-order contrastive loss to further augment the contrast within the joint embedding space. Extensive experiments are conducted on four benchmark datasets under various settings, and the results prove the effectiveness of the proposed method.

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

## A  ADDITIONAL IMPLEMENTATION DETAILS

The source codes of the paper can be downloaded at `https://drive.google.com/file/d/1ANv8rqhWEg6R6wQ7S18jKB2wBA6Arato/view?usp=sharing`. For encoder networks, bidirectional LSTM (Hochreiter & Schmidhuber, 1997; Schuster & Paliwal, 1997) is chosen as the backbone for sequence feature extraction. Following Dong et al. (2022), the training batch size is set to 64, and the length $r$ of the memory queue is set to 1024. During training, we utilize the SGD optimizer with a momentum of 0.9 and a weight decay of 1e-4. The initial learning rate is set to 0.01 and decays by 0.1 at epoch 350. The experiments are conducted on one GeForce RTX 2080 Ti GPU.

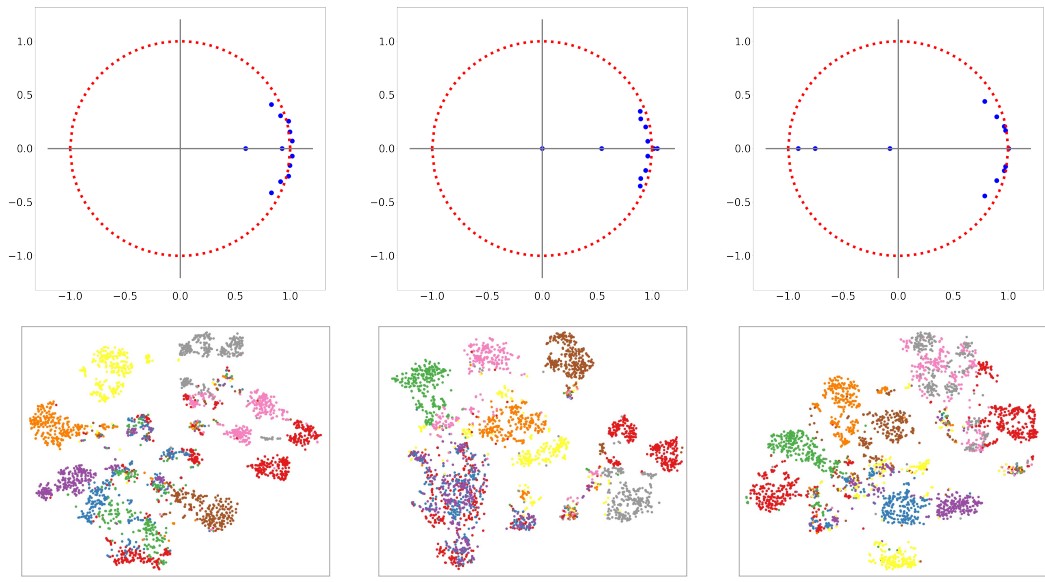

Figure 4: Visualizations of the eigenvalues of the global Koopman dynamics matrix $\mathbf{K}$ on the complex plane (top), as well as the t-SNE visualization of the learned representation (bottom). Best viewed in color.

## B   A DETAILED INTRODUCTION ON KOOPMAN THEORY

The main idea of Koopman theory is to lift the original state space into some higher-dimensional Hilbert space, where the evolution of the states turns to be linear. Formally, given a discrete-time dynamical system,

$$y_{t+1} = F(y_t), \ \ Y \in \mathbb{R}^d. \tag{14}$$

The real-valued observable function $\phi$ is defined on states $y_t$ as $x_t = \phi(y_t)$. The Koopman operator $\mathcal{K}$ acts on the observable $\phi$, which is defined as

$$(\mathcal{K}\phi)(y_t) = (\phi \circ F)(y_t) = \phi(F(y_t)) = \phi(y_{t+1}). \tag{15}$$

In practice, the Koopman operator $\mathcal{K}$ is usually approximated by a finite-dimensional matrix $\mathbf{K}$. This yields a linearly-evolved subspace where the dynamic is governed by the Koopman matrix $\mathbf{K}$:

$$x_{t+1} \ \ = \ \ \phi(y_{t+1}) = \phi(F(y_t)) = \mathbf{K}\phi(y_t) = \mathbf{K}x_t. \tag{16}$$

The Koopman operator can be obtained in a data-driven manner. A widely adopted approach for obtaining a finite-dimensional approximation of the Koopman operator is achieved via Dynamic Mode Decomposition (DMD), which is formulated as:

$$\mathbf{K} = \arg\min_{\mathbf{K}} \|\mathbf{X}_{2:T} - \mathbf{K}\mathbf{X}_{1:T-1}\|_F = \mathbf{X}_{2:T}\mathbf{X}_{1:T-1}^{\dagger}, \tag{17}$$

where $\|\cdot\|_F$ denotes the Frobenius norm and $^{\dagger}$ is matrix pseudo-inverse. $\mathbf{X}_{i:j}$ stands for $[x_i, \cdots, x_j]$.

In recent years, there has been a growing interest in bridging deep networks and Koopman theory by representing a nonlinear-to-linear mapping via neural networks. Such a framework can be effectively optimized in an end-to-end fashion and is well-suited for handling large-scale datasets. For instance, Lusch et al. (2018) propose an auto-encoder framework that embeds the dynamics on a low-dimensional manifold, and identifies Koopman eigenfunctions with three specifically designed loss functions.

## C   ADDITIONAL VISUALIZATIONS

To provide further evidence of the effectiveness of the proposed KTCL method, we conducted additional visualizations, which are presented in Figure 4. As shown, the eigenvalues of the Koopman

dynamics matrices fall near the unit circle, indicating the stability and non-decaying nature of the learned dynamics. Furthermore, the t-SNE visualization of the embedded data confirms the discriminative nature of the learned action sequence representation, thereby facilitating the classification task.

