# OpenReview forum: "Localized Linear Temporal Dynamics for Self-supervised Skeleton Action Recognition"
_ICLR.cc/2024/Conference — ICLR 2024 Conference Withdrawn Submission_

### Official Review · Reviewer_7Efe · 2023-10-26

**Soundness:** 3 good
**Presentation:** 3 good
**Contribution:** 3 good
**Rating:** 5
**Confidence:** 4

**Summary:**

The authors focus on the self-supervised skeleton action recognition and argue that existing methods overlook the 'high-order' temporal contents despite utilizing the spatial-temporal backbone. To remedy this issue, the Koopman Theory is introduced, which can map the complex nonlinear dynamics into linear evolutions by finite-dimensional approximation approach. Then, the 'cross-order' representation considering both dynamic equation and initial value is extracted by forcing the 'first-order'(plain features outputted by typical backbone networks) and 'high-order' feature vectors together. Experimental results adequately verify the effectiveness and robustness of KTCL as well as its components.

**Strengths:**

This paper may have the following strengths worth being mentioned:
a) The proposed method can reach a new state-of-the-art compared with competing methods.
b) The experiments are well organized.

**Weaknesses:**

My main concerns focus on the novelty and fair comparison:
a) The motivation for combining self-supervised learning with Koopman theory for skeletal action recognition may need to be stronger. In other words, deep Koopman learning should exhibit consistent superiority for both self-supervised and supervised frameworks if it robustly works well for action recognition.
b) Regarding deep Koopman learning, the proposed KTCL seems to make no significant difference from others mentioned in the related work section. Undoubtedly, the application innovation accompanied by structural modifications should be encouraged, but the deeper and further improvements may be more attractive for publication.
c) The reviewer is unsure whether the query/key mapping in Figure 2 will bring more training parameters and thus affect the fairness of the performances.

**Questions:**

Thanks for letting the reviewer know about Koopman theory.
To the reviewer's knowledge, it may work like the typical Kernel method conceptually, i.e., projecting the indivisible low-dimensional features into the high-dimension space to make them linearly separable.
According to traditional experiences, deep nonlinear networks may achieve the same target to a large extent.
So, the reviewer is unsure whether the deep spatiotemporal network can fit the Koopman approximation well, as illustrated by Figure 2.
If it does, the motivation for introducing the Koopman approximation may still need to be determined in the manuscript.
Otherwise, why can the method learn a joint embedding space, as shown in Figure 2?
On the other hand, the definitions of 'first order' and 'high order' dynamics may need further clarification.

---

### Official Review · Reviewer_mmWt · 2023-10-28

**Soundness:** 2 fair
**Presentation:** 2 fair
**Contribution:** 2 fair
**Rating:** 3
**Confidence:** 3

**Summary:**

This paper proposes a model that uses the introduced Koopman Temporal Contrastive Learning (KTCL) for self-supervised skeletal action recognition.

The KTCL is inspired by the Koopman theory that can be used to learn discriminative linear system dynamics.

For each action sequence, they first map it into a new space so that the temporal evolution is linear, and later a dynamics-oriented contrastive loss is used to force the dynamics of positive (or negative) samples more similar (or dissimilar).

They further introduce a cross-order contrastive loss to amplify the effect of contrastive across features of different orders.

They show the proposed model achieves better performance on several benchmarks.

**Strengths:**

- The authors conduct many experiments to show the effectiveness of the proposed model.

- The cross-order contrastive loss is quite interesting, but the experimental evaluations and comparisons are quite limited.

- In general, the paper is technical sound, and the organisation of the paper is good.

**Weaknesses:**

Major:

- Using the Koopman operator for the linearisation of non-linear dynamics systems has been explored for many years and not very new, and its application to skeleton-based action recognition is also not new. (i) The newest work, e.g., [A], uses the same idea for skeletal action recognition and further extended to one-shot learning as well. What are the novelties of the proposed model compared to the existing methods in the literature? (ii) It looks like an application of Koopman operator to solve time series / skeleton sequence classification problems.

- Lack of enough literature review of related works that are closely related to the proposed model, for example, the existing methods for handling the temporal information, e.g., bilinear pooling, second-order pooling, covariance pooling, the use of Dynamic time warping, etc.

- A research paper should focus on describing their own innovation. This paper has made a lot of introductions in the background, which are all published works. This is inappropriate. The author of the paper should compress the background content and save the space to describe the methods proposed. For example, Sec. 3.1 of the paper looks almost identical to the Sec. 3.1 of paper [A]! I know these are basics of Koopman theory, but where is your cutting-edge innovations?

- The authors presented Fig. 1 & 2, but these two figures are not properly managed and unable to deliver the core innovations of the proposed method. For example, in Fig. 2, the embedding of features in the joint embedding space is not discussed clearly in either the figure caption or the texts. Figure 3 shows some nice visualisations; however, in Fig. 3(b), what does each different colour mean and how to know if the proposed model works better, and how to compare? At lease some t-SNEs for either baseline models or variants are required to ensure a comparison to support the claim made '... the model has learned discriminative features ...'.

- It is suggested to have a notation section for the maths symbols used in the paper to make them clearer to readers. For example, regular fonts are scalars; vectors are denoted by lowercase boldface letters, matrices by the uppercase boldface, etc. (i) Eq. (1) What is F, Y and t? These concepts should be mentioned clearly to readers if you show them in the paper. (ii) What are the dimensions of  $x_i$ and $\textbf{X}$?

- Lack of experiments to show the effectiveness w.r.t. baselines etc: (i) Sec. 3.2, the authors mentioned that their method is built on the basic structure of MoCo, and they cite two works; however, these methods are not evaluated or compared in the paper as e.g., baselines, etc. Such comparisons are also important to show the effectiveness of the proposed model on top of existing models like MoCo. (ii) The authors also use the first-order feature representation combined with the high-order dynamics; however, there is no evidence to show the performance gain is from the high-order dynamics or the first-order representations from the base models, etc.

- The reasons for the underperforming cases are not even mentioned in the paper, this is a noteworthy shortcoming. These cases should be discussed and analysed thoroughly, e.g., the method is based on HiCo (AAAI'23), but it achieves 74.5% on the X-Set of NTU-RGB+D 120, but your extension (with KTCL) of HiCo only achieves 74.1%, this is a performance drop that should be mentioned and discussed in the paper, as to readers, these are very important. Again, in Table 2, why existing method such as SkeAttnCLR (IJCAI'23) achieves 69.4% on the X-Sub of NTU-RGB+D 60, but the proposed KTCL only achieves 67.4%, any discussions why the performance drops ~ 2%?

[A] Neural Koopman Pooling: Control-Inspired Temporal Dynamics Encoding for Skeleton-Based Action Recognition, CVPR'23

Minor:

- The authors say 'the similarity between the embeddings is measured as the dot product', is the features being normalised (e.g., cosine similarity) or the magnitudes of feature representations are considered as well?

- How you obtain the skeleton sequence to form each T-frame? These information should be clear to readers, if similar to existing works, then related citations should be there clearly.

- Eq. (12) $m_j$ is not explained, what is its relationship w.r.t. $m_j^d$, $m_j^t$, and $m_j^s$. A concatenation similar to the obtaining of
$z$?

**Questions:**

The reviewer observed that this is a withdrawn submission from NeurIPS'23, where I also served as one of the reviewers.

After a thorough reading of the paper, the reviewer noted that there are no significant updates compared to the original version (and all reviewers' comments are generally ignored and concerns remained unsolved).

The reviewer would like to emphasize once more that, 'Note that reviewers’ comments indeed help the improvement of the quality of research, and if issues/problems are not being well addressed, it will be surely again raised in the next round of review. I suggest the authors take all reviewers’ comments into consideration as that is path to high quality works.' (copied directly from my previous NeurIPS'23 suggestions)

Please refer to the 'Weaknesses' section for questions, issues, or concerns that need to be addressed.

---

### Official Review · Reviewer_qrLh · 2023-10-30

**Soundness:** 3 good
**Presentation:** 3 good
**Contribution:** 2 fair
**Rating:** 5
**Confidence:** 3

**Summary:**

The paper addresses the challenge of self-supervised skeleton action recognition. The authors employ a dynamics-oriented contrastive loss grounded in Koopman Theory, which translates complex dynamic systems into linear ones. Their methodology can be categorized along three dimensions:
- Leveraging contrastive learning based on global temporal dynamics,
- Integrating the proposed loss at the segment level,
- Applying infoNCE in temporal and spatial dimension, respectively, by pairng higher-order dynamics feature and first-order dynamics feature.

They validated their approach using the NTU-60, NTU-120, and PKU-MMD datasets.

**Strengths:**

- The authors presented a contrastive loss derived from Koopman theory, enabling the model to discern higher-order temporal evolution patterns. Applying dynamics theory to the representation of human action recognition is interesting.
- The paper is well-structured with clear and coherent writing.
- The outcomes demonstrate state-of-the-art performance and can be replicated by their code.

**Weaknesses:**

While the approach presented is innovative and theoretically sound, the experiments and analyses don't provide a compelling argument for several reasons:

- In Figure 3-(a) and Figure 4, only four samples of the eigenvalues from the Koopman dynamics matrix in total are visualized. Utilizing statistics might offer a more robust argument for the stability of the learned linear dynamics.
- The t-SNE representation in Figure 3-(b) illustrates that the embedding spaces are well-categorized. However, its persuasiveness would be enhanced if compared with t-SNE representations of baseline models.
- I would advise the authors to conduct ablation studies for local-level dynamics and experimenting with variations in the value of $s$

**Questions:**

- What happens if $s$ increases and decreases?
- Can there be a way to visualize the evolution pattern of the actions?

**Details Of Ethics Concerns:**

I have no concerns regarding the ethics.

---

### Official Review · Reviewer_A4dc · 2023-10-31

**Soundness:** 2 fair
**Presentation:** 2 fair
**Contribution:** 2 fair
**Rating:** 3
**Confidence:** 5

**Summary:**

This paper proposes a model that uses the introduced Koopman Temporal Contrastive Learning (KTCL) for self-supervised skeletal action recognition. Drawing inspiration from the Koopman theory, which facilitates the learning of distinguishing linear system dynamics The model integrates a cross-order contrastive loss. This enhancement magnifies the contrastive effect across multi-order features. Additionally, a segment-level KTCL has been incorporated to understand dynamics at the segment scale. The proposed model achieves better performance on 4 benchmarks.

**Strengths:**

The proposed cross-order contrastive loss is new and interesting.

The authors conduct many experiments to show the effectiveness of the proposed model.

This paper is well-written and the organization of the paper is good.

**Weaknesses:**

While the application of the Koopman operator for linearizing non-linear dynamics is not entirely new, nor is its use in skeleton-based action recognition, there are notable questions to consider. (i) With the recent study by 'Wang et al. (2023)' applying similar principles to skeletal action recognition and extending it to one-shot learning, what distinct advancements does the proposed model offer compared to existing methods? (ii) At a glance, this seems to be another application of the Koopman operator targeting time series or skeletal sequence classifications.

Regarding the 'SEGMENT-LEVEL LOCALIZED LINEAR DYNAMIC', it appears to be a segment-specific adaptation of the KTCL. How does this qualify as 'cross-matching'?

Given that the proposed approach draws on the encoder from HiCo, the observed performance enhancement on expansive datasets like NTU 60 and NTU 120 seems relatively modest. What accounts for this limited improvement?

Many contrastive learning approaches for skeletons primarily adopt CrossCLR (the pioneering work) as their backbone structure. What motivated the choice to utilize the more advanced HiCo in this work?

**Questions:**

Please refer to the weakness